# Clustering using Approximate Nearest Neighbour Oracles

**Enayat Ullah**                                                  *enayat@jhu.edu*
*Department of Computer Science*
*Johns Hopkins University*

**Harry Lang**                                                  *hlang08@gmail.com*
*Unafffliated*

**Raman Arora**                                              *arora@cs.jhu.edu*
*Department of Computer Science*
*Johns Hopkins University*

**Vladimir Braverman**                                        *vb21@rice.edu*
*Department of Computer Science*
*Rice University*

**Reviewed on OpenReview:** *https://openreview.net/forum?id=TzRXyO3CzX*

## Abstract

We study the problem of clustering data points in a streaming setting when one has access to the geometry of the space only via approximate nearest neighbour (ANN) oracles. In this setting, we present algorithms for streaming $O(1)$-approximate $k$-median clustering and its (streaming) coreset construction. In certain domains of interest, such as spaces with constant expansion, our algorithms improve upon the best-known runtime of both these problems. Furthermore, our results extend to cost functions satisfying the approximate triangle inequality, which subsumes $k$-means clustering and $M$-estimators. Finally, we run experiments on Census1990 dataset wherein the results empirically support our theory.

## 1 Introduction

In recent years, the *streaming* model of computation has become very popular for analyzing large datasets. In this model, the objective is to compute some statistic or function of the dataset (eg: frequent items, norms, etc.), with access to data points sequentially. Moreover, only one or a few passes through the stream are allowed and the algorithm is permitted to use memory, *sub-linear* in the number of data points. A general technique is to compute and store a succinct representation, typically called a *sketch*, of the points seen till a given point in the stream and to update it efficiently upon seeing a new point. A key challenge is to ensure that the size of the sketch is sub-linear in the number of data points. For more details, we refer the reader to Muthukrishnan et al. (2005); Babcock et al. (2002) for comprehensive surveys.

In this work, we are concerned with the problem of clustering over data steams using only *approximate nearest neighbour* (ANN) search. Clustering is a ubiquitous unsupervised data processing and exploratory technique wherein the high-level goal is to accumulate *similar* points together. Clustering and ANN search are two classical geometric problems which have been extensively studied and it is folklore that these two bear connections. We investigate this connection with the following questions: *"Can we devise clustering algorithms with provable guarantees with access to the geometry of the space via only ANN queries"?* Secondly, *"are there (interesting) settings wherein this yields an overall improved runtime compared to prior work"?*

To this end, we present algorithms for $k$-median clustering which work with *blackbox* ANN oracle accesses. A related problem is that of building *coresets*, which are small but *representative* sub-sample of the dataset such that it approximates the cost of *every* solution (see Section 2 for a formal definition). In this work,

we are concerned with building coresets with respect to the $k$-median clustering cost. For both these problems, our algorithms are simple and provably correct adaptations of known algorithms: Braverman et al. (2011) for streaming $k$-median clustering, and both Chen (2009) and Braverman et al. (2016) for its coreset constructions, to the ANN oracle setting - we will call these *Algorithm 1*, *Algorithm 2* and *Algorithm 3* respectively. We furthermore instantiate these with specific ANN constructions, which aided by fast ANN search, give improvements in overall runtime. In practice, it is observed that many point sets in high-dimensional ambient space usually have an intrinsic low-expansion structure. Motivated by this, there has been a series of results studying ANN search in *small expansion* spaces (to be formalized later), for example (Beygelzimer et al., 2006; Krauthgamer and Lee, 2004). We show that in these constant expansion spaces we get a *strict* improvement in terms of running time, of $k$-median clustering and its coreset construction. Specifically, with $n$ points and $k$ target clusters, the runtime improves from $\mathcal{O}(nk \log n)$ to $\mathcal{O}(n \log(k \log(n/k)))$ for both problems without affecting the space complexity. Moreover, in domains other than constant expansion spaces, popular methods like Locally Sensitive Hashing (LSH) (Indyk et al., 1997), when used as ANN oracles, can trade-off *faster* runtime for *worse* space complexity, in clustering.

We summarize our main contributions as follows.

1. We present algorithms for streaming $k$-median clustering and its coreset construction using only blackbox ANN oracle accesses.

2. Equipped with sub-linear time ANN search, this speeds up the overall runtime of streaming $k$-median clustering and its coreset construction. In particular, in spaces with constant *expansion constant*, the runtime improves from $O(nk \log(n))$ to $O(n \log(k \log(n/k)))$ without sacrificing the space complexity.

3. A minor and unrelated result is that we improve the space complexity of streaming $k$-median clustering algorithm of Braverman et al. (2011) from $O(k \log(n))$ to $O(k \log(n/k))$ (with and without using ANN). We note that this improvement only manifests in certain restrictive regimes.

4. Our results extend to distance functions satisfying approximate triangle inequality, such as those used in $k$-means clustering, $M$-estimators etc.

## 1.1 Prior Work

The problems of $k$-median and $k$-means clustering have been widely studied. The rich history consists of algorithms with provable guarantees as well as heuristics which work well in practice. One popular such heuristic for $k$-means clustering, based on local search, is Lloyd's heuristic (Lloyd, 1982), which alternatively updates the cluster assignments and the cluster centers. However, it is easy to come up with arbitrarily bad examples for it. The first polynomial time algorithm which gives a constant approximation was given by Charikar et al. (1999), which was then improved by a line of work (Charikar and Guha, 2005; Jain and Vazirani, 2001; Arya et al., 2004). The first result for $k$-median clustering in the streaming setting was given by Guha et al. (2000) which achieves a $O(2^{1/\epsilon})$ approximation using $O(n^{\epsilon})$ space. Meyerson (2001) introduced the related problem of Online Facility Location (OFL) which paved way for a stream of works on $k$-median clustering using OFL. In particular, Charikar et al. (1999) gave an algorithm that produces a $\mathcal{O}(1)$ approximate solution using $O(k \log^2 n)$ space, subsequently improved to $O(k \log n)$ by Braverman et al. (2011). Another popular technique for the clustering problem is to construct $\epsilon$-coresets of the dataset with respect to the clustering cost. Herein, the idea is to extract a weighted set from the original set such that the cost of *any* $k$-median clustering on this set is *close* to that on the original set. The first coreset construction was given by Har-Peled and Mazumdar (2004) for $k$-median and $k$-means clustering which uses $O(k\epsilon^{-d} \log^{2d+2} n)$ space. Subsequent works by Frahling and Sohler (2005); Chen (2009); Feldman and Langberg (2011); Braverman et al. (2016) improved upon the guarantees and also showed how to adapt the construction to the streaming setting.

We note that there are indeed previous works which explore this connection between clustering and ANN search, however most of these are heuristics which use ANN as an implementation detail (Deng and Zhao, 2017; Franti et al., 2006). The most related to ours is that of Shindler et al. (2011), which as ours, builds

upon the work of Braverman et al. (2011) for streaming $k$-median clustering. Their work addresses its application to real data sets, and one of the *tricks* they employ is ANN search, which they remark enjoys theoretical guarantees as well. However, they are primarily limited to only one such ANN method - random projections, which they use in their implementation. Moreover, they only give bounds in expectation, their runtime has an undesirable dependence on optimal cost, and suffer from a large unspecified constant in their approximation guarantee. On the other hand, our methods work with blackbox ANN oracles and we give high probability bounds with precise constants.

## 1.2 Overview of results

In this section, we state our results and discuss the key insights. The reader is encouraged to refer to section 2 to see exact definitions of the technical terms used. Our results are presented in a general setup wherein one has access to the geometry of the set via blackbox ANN oracles. Our theorems are therefore stated in terms of the *update*, *query* and *space* complexities of the ANN oracle/data structure, which are functions of its size (i.e. number of data points). We first state the theorem for streaming $k$-median clustering.

**Theorem 1** ($k$-median clustering). *Given an ANN oracle with $q(\cdot)$ query time, $u(\cdot)$ update time and $s(\cdot)$ space complexity, a $[O(1), O(1)]$-bicriterion approximate $k$-median clustering solution can be computed on a stream of $n$ points with probability at least $1 - 1/poly(n)$ while using $s(k \log(n/k))$ space and with $O(u(k \log(n/k)) + q(k \log(n/k)))$ worst-case update time.*

Even though the above theorem is stated for finding a bicriterion solution, we will, in section 3, show how to efficiently convert it into an $O(1)$-approximate solution. In terms of runtime, our algorithms makes *exactly* $n$ calls to the ANN oracle for $n$ points. We remark that this is *optimal* in terms of oracle complexity in our setting - i.e when we have access to the geometry via ANN oracles. It is easy to see as a single *bad* point can arbitrarily change the optimal clustering, and therefore we need to query at every point. Another (minor) implication of this theorem is a separate (unrelated to ANN) result: we improve the space complexity for $k$-median clustering of Braverman et al. (2011) from $O(k \log(n))$ to $O(k \log(n/k))$ even without ANN oracles. A final remark is about computational complexity of $k$-median clustering. For $n$ points and $k$ clusters, the classic lower bound on runtime of any $O(1)$-approximate randomized or deterministic algorithm is $\Omega(nk)$ (Guha et al., 2000; Mettu and Plaxton, 2004). However, this lower bound is established in a limited computational framework that counts the number of *distance queries* any algorithm necessarily needs to make. Our result indicates that it is quite pessimistic, as there are spaces that admit sub-linear time ANN search (like Euclidean spaces), which suffices for solving the problem of $k$-median clustering *faster*.

We now present our result on constructing $\epsilon$-coresets.

**Theorem 2** (Coreset). *Given an ANN oracle with $q(\cdot)$ query time, $u(\cdot)$ update time and $s(\cdot)$ space complexity, an $\epsilon$-coreset for $k$-median can be computed on a stream of $n$ points with probability at least $1 - 1/poly(n)$ in $O(u(k \log(n)) + q(k \log(n)))$ worst-case update time and $O(\epsilon^{-2} k \log^2 n + s(k \log(n)))$ space.*

We adapt two coreset construction frameworks: Chen (2009) (*Algorithm 2*) and Braverman et al. (2016) (*Algorithm 3*), to the ANN setting. Equipped with fast ANN search, this similarly allows us to improve their runtimes. For the first, we adapt the ring construction of Chen (2009) to the ANN setting, and show that this approximation preserves the theoretical properties. For the second, we employ techniques from Braverman et al. (2016) which allows construction of coresets by importance sampling even when sampling probabilities can be *overestimated*. We essentially show how using ANN oracles can lead to *controllable* worsening of sampling probabilities, and still enable us to construct $\epsilon$-coresets. An important remark is that second framework uses the $O(1)$ approximate $k$-median clustering algorithm of Braverman et al. (2011) as a crucial subroutine. Therefore our *Algorithm 1* which adapts the $k$-median clustering algorithm of Braverman et al. (2011) to the ANN oracle setting is important to make the coreset construction *Algorithm 3* wholly implementable using ANN oracles.

**Applications.** We discuss how specific ANN oracle instantiations manifest in overall runtime and space bounds. Table 1 gives the query complexity of various ANN methods, as well as space and time complexities when used for $k$-median clustering and $\epsilon$-coresets. For coreset construction, for ease of exposition, we only list bounds obtained from adapting the algorithm of Braverman et al. (2016) (*Algorithm 3*), and omit Chen

| ANN method | Geometry of the set | $q(m)$-Query complexity | - | $k$-median clustering | $\epsilon$-coreset |
|---|---|---|---|---|---|
| Cover trees | Expansion | $O(\tau^{12} \log m)$ | time | $O(n \log(\tilde{k}))$ | $O(n \log(\bar{k}))$ |
| | Constant $\tau$ | | space | $O(k \log(n/k))$ | $O(\epsilon^{-2} k \log^2(n))$ |
| Navigating Nets | Expansion | $\tau^{O(1)} \log m$ | time | $O(n \log \tilde{k})$ | $O(n \log \bar{k})$ |
| | Constant $\tau$ | | space | $O(k \log(n/k))$ | $O(\epsilon^{-2} k \log^2(n))$ |
| LSH | $(\mathbb{R}^d, l_1)$ | $O(m^{1/2} d)$ | time | $O(nd\sqrt{\tilde{k}})$ | $O(nd\sqrt{\bar{k}})$ |
| | | | space | $O(\tilde{k}^{3/2} + \tilde{k}d)$ | $O(\epsilon^{-2} \log n (\bar{k}^{3/2} + \bar{k}d))$ |
| LSH | Hamming $(\{0,1\}^d, l_1)$ | $O(m^{1/2} d)$ | time | $O(nd\sqrt{\tilde{k}})$ | $O(nd\sqrt{\bar{k}})$ |
| | | | space | $O(\tilde{k}^{3/2} + \tilde{k}d)$ | $O(\epsilon^{-2} \log n (\bar{k}^{3/2} + \bar{k}d))$ |
| Data-dependent partition | $(\mathbb{R}^d, \ell_2)$ | $O(m^{1/7} d)$ | time | $O(nd\tilde{k}^{1/7})$ | $O(nd\bar{k}^{1/7})$ |
| | | | space | $O(\tilde{k}^{8/7} + \tilde{k}d)$ | $O(\epsilon^{-2} \log n (\bar{k}^{8/7} + \bar{k}d))$ |
| Exhaustive Search | $(\mathcal{X}, \rho)$ Any metric space | $O(m)$ | time | $O(nk \log(n/k))$ | $O(nk \log(n))$ |
| | | | space | $O(k \log(n/k))$ | $O(\epsilon^{-2} k \log^2(n))$ |

Table 1: Runtime and space complexity using different ANN methods in various domains. The query complexity $q(m)$ of an ANN method is the time complexity to compute an ANN from $m$ candidates. For space partitioning methods (LSH and others), the query complexity is $\mathcal{O}(m^\varrho d)$, where $\varrho$ usually is poly$(1/c)$, $c$ being the approximation factor of ANN. For example, in Euclidean space, we get $\varrho = 1/c^2$. For simplicity of presentation, we present the results with $c = 2$, and give bounds in terms of $\tilde{k}$ and $\bar{k}$ where $\tilde{k} := k \log(n/k), \bar{k} := k \log n$.

(2009)(*Algorithm 2*). In the table, exhaustive search denotes the brute force nearest neighbour search method which serves as a baseline. Furthermore, we define the terms $\tilde{k}$ and $\bar{k}$ as $\tilde{k} := k \log(n/k), \bar{k} := k \log n$, which are used in the table. We see that LSH based methods have faster runtimes but they have worse space complexity. On the other hand, in constant expansion spaces, cover trees and navigating nets give fast runtimes with the same space complexity, which establishes out claimed improvement in runtime.

## 2 Preliminaries and definitions

In this section, we formalize the problem and provide key definitions. Let $(\mathcal{X}, d)$ be a metric space, and let $P \subseteq \mathcal{X}, |P| = n$ be the set of points observed in a stream. Given integer $k \in [1, n]$, the desired number of clusters, the problem of $k$-median clustering asks for a set of $k$ points called *cluster centers* (or just centers) from $\mathcal{X}$ such that the sum of distances of points in $P$ to the nearest point in $\mathcal{C}$ is minimized; therefore it can be posed as the following optimization problem.

$$\min_{\mathcal{C} \subseteq \mathcal{X}, |\mathcal{C}| = k} \sum_{p \in P} \min_{c \in \mathcal{C}} d(p, c) \qquad \text{(k-median clustering)}$$

For a fixed $k$ and set of points $P$, let $\mathsf{OPT}(P, k)$ denote the cost incurred by the optimal selection of $k$ centers from $P$. A related problem is Online Facility location (OFL), which is used in our algorithm.

**Online Facility Location.** Given a fixed facility cost $f > 0$, the *facility location* problem relaxes the constraint of selecting exactly $k$ centers to adding a penalty cost of $f$ per center to the total cost. Intuitively,

the set of points $P$, called *demands*, are to be *served* via a set of *facilities* such that the total cost of serving the demands via the nearest facility plus opening facilities is small. *Service cost* of a point $p \in P$ is the cost of *serving* this demand via the nearest facility, defined via its distance to the nearest facility. Therefore the total service cost is $\sum_{p \in P} \min_{c \in \mathcal{C}} d(p, c)$. *Facility cost* is the cost incurred in opening facilities; the total facility cost $= |\mathcal{C}| f$. Therefore, facility location problem can be formulated as : $\min_{\mathcal{C} \subseteq \mathcal{X}} \sum_{p \in P} \min_{c \in \mathcal{C}} d(p, c) + |\mathcal{C}| f$.

The *online* variant of facility location, known as *Online Facility Location* (OFL) considers the problem of facility location in the streaming setting with the added constraint that once a point is assigned to a particular facility, it cannot be changed in the future (even if we open a nearer facility). Moreover, once you designate a point as a facility, it remains a facility henceforth. Equivalently, the service and facility costs incurred while observing the stream are irrevocable.

**$\alpha$-metric Spaces and $k$-service clustering.** The problem of *$k$-service clustering* generalizes the objective of $k$-median clustering to $\alpha$-metric spaces, which are defined by relaxing the triangle inequality to what is called $\alpha$-approximate triangle inequality (ATI): $\rho(x, y) \leq \alpha(\rho(x, z) + \rho(z, y)) \ \forall \ x, y, z \in \mathcal{X}$ (see Appendix A for formal definitions). It can be posed as the following optimization problem.

$$\min_{\mathcal{C} \subseteq \mathcal{X}, |\mathcal{C}| = k} \sum_{p \in P} \min_{c \in \mathcal{C}} \rho(p, c) \qquad \text{(k-service clustering)}$$

We note that $k$-median and $k$-means clustering are special cases of the above (see Appendix A). We now define some notions of approximations which are widely used for clustering problems.

**Definition 1** ($\lambda$-approximation). *Let $k \in \mathbb{N}, \lambda > 0$. For the $k$-service clustering problem, a $\lambda$-approximation is a tuple $(\mathcal{C}, u)$ where $\mathcal{C} \subseteq \mathcal{X}$ & $u : \mathcal{X} \to [0, \infty]$ such that $\sum_{p \in P} \min_{c \in \mathcal{C}} u(c)\rho(c, p) \leq \lambda OPT(P, k)$ & $|\mathcal{C}| = k$.*

In the above, we are tolerating a constant approximation to the optimal cost but returning a set of exactly $k$ centers. A bicriterion approximation, defined below, further also relaxes the constraint on the size of the set of centers from $k$ to $O(k)$.

**Definition 2** (($\lambda, \kappa$)-bicriterion approximation). *Let $k \in \mathbb{N}, \lambda > 0$ and $0 < \kappa < 1$. A $(\lambda, \kappa)$-bicriterion approximation is a tuple $(\mathcal{C}, u), \mathcal{C} \subseteq \mathcal{X}$ & $u : \mathcal{X} \to [0, \infty]$ such that $\sum_{p \in P} \min_{c \in \mathcal{C}} u(c)\rho(c, p) \leq \lambda OPT(P, k)$ and $|\mathcal{C}| \leq \kappa k$*

**Coresets.** We first give a formal definition of an $\epsilon$-coreset, and then discuss some applications. A query space over $\mathcal{X}$ is a tuple $(P, w, f, Q)$ where $P \subseteq \mathcal{X}$, $w : P \to [0, \infty]$ is a weight function, $Q$ is the set of all possible queries, $W$ the set of all possible weight functions, and $f : \mathcal{X} \times W \times Q \to [0, \infty]$.

**Definition 3** ($\epsilon$-coreset). *An $\epsilon$-coreset for the query space $(P, w, f, Q)$ is a tuple $(Z, u)$ where $Z \subseteq \mathcal{X}$ and $u : Z \to [0, \infty]$ are points and their corresponding weights respectively such that for every $q \in Q$, we have $(1 - \epsilon)f(P, w, q) \leq f(Z, u, q) \leq (1 + \epsilon)f(P, w, q)$*

In the context of $k$-median clustering, a query is any set of $k$ points i.e. any candidate solution of $k$ centers. Informally, for a given problem, coresets are small, succinct representations of the dataset such that the cost of *any* candidate solution on the coreset is close to that on the original set. Recently, coresets have been designed for various problems such as $k$-median clustering, $M$-estimation, Bayesian inference, in offline and streaming settings (Chen, 2009; Feldman and Langberg, 2011; Braverman et al., 2016; Huggins et al., 2016; Braverman et al., 2019). In this work, we are primarily concerned with the function being the $k$-median cost, but it can easily be extended to $k$-service cost. A useful property of coresets is that they are *closed* under operations like union and composition, formalized as Proposition 1 in Appendix A.

**Approximate Nearest Neighbour Oracles.** Approximate nearest neighbour search is a classical geometric problem concerning with the design of data structures and algorithms which support sub-linear time (in number of points) queries for nearest neighbour search. We now define an approximate nearest neighbour oracle. A more detailed discussion is provided in the appendix.

**Definition 4** (Approximate Nearest Neighbour ($c(\delta)$-ANN) Oracle). *For a given space $\mathcal{X}$, $\alpha$-ATI function $\rho$ and set $\mathcal{C} \subseteq \mathcal{X}$, an algorithm $\mathcal{A}$ is a $c(\delta)$-ANN Oracle if given a query point $x \in \mathcal{X}$, it returns $(y, \rho(x, y))$*

where $y \in \mathcal{C}$, such that with probability at least $1 - \delta$, $\rho(x,y) \leq c(\delta) \cdot \min_{z \in \mathcal{C}} \rho(x,z)$. The query, update and space complexity of ANN is denoted by $q(|\mathcal{C}|), u(|\mathcal{C}|)$ and $s(|\mathcal{C}|)$ respectively.

Cover trees (Beygelzimer et al., 2006), Navigating Nets (Krauthgamer and Lee, 2004) and Locally Sensitive Hashing (LSH) (Indyk et al., 1997; Andoni et al., 2018) are some examples which can serve as an ANN oracle. A key quantity which these bounds depend on is expansion constant. Informally, the expansion constant of a metric space is the largest factor by which the number of points in any ball increases when we *expand* a ball to be of twice the radius.

**Definition 5** (Expansion constant $\tau$)**.** *(Beygelzimer et al., 2006) Let* $(S, \rho)$ *be a metric space. Fix any* $r > 0$, *define* $B_\rho(p,r) = \{x : \rho(p,x) \leq r\}$ *and* $B_\rho^S(p,r) = B_\rho(p,r) \cap S$. *The Expansion Constant of set* $(S, \rho)$ *is defined as* $\tau$, *if* $\tau \geq 2$ *is the smallest number such that* $\left|B_\rho^S(p,2r)\right| \leq \tau \left|B_\rho^S(p,r)\right|$ *for any* $r$.

## 3   Streaming $k$-median Clustering

In this section, we discuss the algorithm for streaming $k$-median clustering (*Algorithm 1*) and sketch its analysis. We remind the reader that in the streaming model of computation, we observe data points one at a time, and the goal is to answer some pre-defined query on the whole stream. Our algorithm builds upon the algorithms of Charikar et al. (2003); Braverman et al. (2011) which uses OFL as a subroutine. The algorithm in full is presented in Algorithm 1. The algorithm runs in phases, wherein each phase corresponds to a *guess*, say, $L$, of the lower bound to the optimal cost OPT. After reading a new point, the algorithm computes its approximate nearest neighbour in the set of candidate facilities, and then either adds the point to the facility set or incurs a service cost - line 8 of Algorithm 1, where the notation *if probability(x) then* $y$ denotes the random event to take action $y$ with probability $x$, by tossing a (Bernoulli) coin with bias $x$. If the guess is too low, we either end up opening too many facilities or the service cost grows large. At this point, we trigger a phase change where we increase our guess by a constant factor $\beta$, and do a *soft* reset. We empty the facility set and push its points to the start of the stream. If we keep incrementing the phase, at some point our guess to the lower bound exceeds the optimal cost OPT. We call this phase the *critical phase*. We show that upon setting the values of parameters appropriately, the algorithm processes the whole stream before the critical phase and returns a constant approximate solution.

There are essentially three changes to the original algorithm of Braverman et al. (2011): line 7, line 13, and the value of the $f$. In line 7, we use ANN instead of exact nearest neighbour and in line 13 we change the condition from $|\mathcal{C}| > (\gamma - 1)k(1 + \log(n))$ to $(\gamma - 1)k(1 + \log(n/k))$. These two changes *independently* improve the runtime and space complexity of Braverman et al. (2011) respectively. The first improvement comes from faster runtime of ANN search compared to exact nearest neighbour computation, and the second comes from an improved analysis of Braverman et al. (2011).

Intermediate to our analysis is the analysis of OFL with ANN oracles. As was shown in Meyerson (2001); Braverman et al. (2011) that the service cost and number of facilities is at most a constant times optimum (using distance oracles), we show that that same holds with ANN oracles. We formalize this in Theorem 3.

**Theorem 3** (OFL)**.** *Given an* $c(\delta)$*-ANN oracle with* $u(\cdot)$ *update,* $q(\cdot)$ *query and* $s(\cdot)$ *space complexity,* OFL *run on a stream of $n$ points from an $\alpha$-metric space with facility cost* $f = \frac{L}{1 + \log n/k}$ *where* $L \leq$ OPT *incurs service cost at most* $\left(3c(\delta/4n)\alpha + \frac{e}{e-1}\left(1 + \log\left(\frac{4}{\delta}\right)\right)\right)$ OPT *and produces at most* $(1 + 6\log(4/\delta)c(\delta/4n)\alpha)k(1 + \log(n/k))$ *facilities, with probability at least* $1 - \delta$. *The algorithm runs in* $\mathcal{O}\left(n(q(k\log(n/k)) + u(k\log(n/k)))\right)$ *time.*

We now give the main theorem for Algorithm 1 with the proof in the appendix.

**Theorem 4.** *Given a stream of $n$ points $P$ from an $\alpha$-metric space $(\mathcal{X}, \rho)$ and a positive integer $k$, Algorithm 1 executed with parameters set as* $\beta = 2\alpha^2 c_{OFL} + 2\alpha$ *and* $\gamma = \max\left\{4\alpha^3 c_{OFL}^2, \beta k_{OFL} + 1\right\}$, *where* $c_{OFL} := \left(3c(\delta/4n)\alpha + \frac{e}{e-1}\left(1 + \log\left(\frac{4}{\delta}\right)\right)\right)$, $k_{OFL} := (1 + 6\log(4/\delta)c(\delta/4b)\alpha)$ *returns a set $\mathcal{C}$ with* $|C| = \mathcal{O}\left(k\log(n/k)\right)$ *such that* COST$(P, \mathcal{C}) \leq \frac{\alpha\beta\gamma}{\beta - \alpha}$OPT$(P,k)$ *with probability at least* $1 - \delta$. *The algorithm runs in time* $\mathcal{O}\left(n(q(k\log(n/k) + u(k\log(n/k)))\right)$ *and uses space* $s(k\log(n/k))$.

---

**Algorithm 1** Streaming $k$-median clustering

---

1: **Input:** Integer $k \geq 1$, stream $P$ of $n$ points, a $c(\delta)$-approximate ANN Oracle
2: $L_1 \leftarrow 1, i \leftarrow 1, w(p) \leftarrow 1$ for all points $p \in P$
3: **while** solution not found **do**
4:      $\mathcal{C} \leftarrow \emptyset, \mathsf{COST} \leftarrow 0, f \leftarrow L_i/(k(1 + \log(n/k)))$
5:      **while** stream not ended **do**
6:          $x \leftarrow$ next point from the stream
7:          $(y, \rho(x,y)) \leftarrow$ ANN Oracle$(x, \mathcal{C})$
8:          **if** probability$\left(\min\left(\frac{w(x)\rho(x,y)}{f}, 1\right)\right)$ **then**
9:             $\mathcal{C} \leftarrow \mathcal{C} \cup \{x\}$
10:          **else**
11:             $\mathsf{COST} \leftarrow \mathsf{COST} + w(x) \cdot \rho(x,y)$
12:             $w(y) \leftarrow w(y) + w(x)$
13:          **if** $\mathsf{COST} > \gamma L_i$ or $|\mathcal{C}| > (\gamma - 1)k(1 + \log(n/k))$ **then**
14:             Break and raise flag
15:      **if** flag raised **then**
16:          Push facilities in $\mathcal{C}$ to the start of stream
17:          $L_{i+1} \leftarrow \beta L_i, \ i \leftarrow i + 1$
18:      **else** Declare solution found
19: **Output:** $\mathcal{C}, \mathsf{COST}$

---

**Constant expansion spaces.** An immediate corollary of the above result is obtained by using cover tree for fast ANN search in spaces of constant expansion, yielding faster runtime.

**Corollary 1.** *Given a stream of $n$ points $P$ from an $\alpha$-metric space $(\mathcal{X}, \rho)$ with constant expansion, and a positive integer $k$, Algorithm 1 executed with cover tree data structure as the ANN oracle, and with parameters specified in Theorem 1 returns a set $\mathcal{C}$ with $|C| = \mathcal{O}\left(k \log(n/k)\right)$ such that $\mathsf{COST}(P, \mathcal{C}) \leq \frac{\alpha\beta\gamma}{\beta - \alpha} \mathsf{OPT}(P, k)$ with probability at least $1 - \delta$. The algorithm runs in time $\mathcal{O}\left(n \log\left(k \log\left(n/k\right)\right)\right)$ and uses space $O(k \log(n/k))$.*

**Extensions:** We remark that in the worst case, Algorithm 1 executes in $O(\log_\beta \mathsf{OPT})$ phases. So the overall runtime has an undesirable dependence on $\mathsf{OPT}$. Moreover, the size of the cluster set produced is $\tilde{O}(k)$ and not exactly $k$. Below, we discuss how the algorithm can be provably tweaked to remove these discrepancies.

**1. Improved runtime via a routine from Lang (2017).** To remove this dependence on $\mathsf{OPT}$, Braverman et al. (2011) proposed a *pruning* method, but it works only for average-case update time. Instead, we use the deterministic $\mathcal{B}(X, k)$ routine of Lang (2017), which not only gives worst case update time guarantee but also takes advantage of the ANN setting. Given $|X| = n$ points and a positive integer $k$, $\mathcal{B}(X, k)$ returns a weighted set $Z$ of size $\left\lceil \frac{n+k}{2} \right\rceil$ such that $\mathsf{COST}(X, Z) \leq 2\mathsf{OPT}(X, k)$. Incidentally, if the nearest neighbour graph of $X$ is pre-computed, the runtime of $\mathcal{B}(X, k)$ is $O(n)$. If $\mathcal{B}(X, k)$ is executed using a $c(\delta)$ ANN oracle, the set $Z$ returned satisfies $\mathsf{COST}(X, Z) \leq 2c(\delta)\mathsf{OPT}(X, k)$ with probability at least $1 - \delta$ and runs in time $O(n \, q(n))$. After every phase change, we take the union of the previous and new facilities $K$ and call $\mathcal{B}(K, k)$ on this set. Since in each phase, we get $O(k \log(n/k))$ facilities, the number of facilities removed by $\mathcal{B}(K, k)$ is also $O(k \log(n/k))$. In this way, we are ensured to read $O(k \log(n/k))$ new points at every phase, which bounds the number of phases by $O(n/(k \log(n/k)))$. Therefore, executing $\mathcal{B}(K, k)$ via ANN oracles, the total time is $O(n(q(k \log(n/k)) + u(k \log(n/k)))$.

**2. Extracting exactly $k$ points.** As pointed out before, Algorithm 1 produces $O(k \log(n/k))$ clusters. To reduce it to exact $k$ points, we can use any Random Access Machine (RAM) algorithm, for example Mettu and Plaxton (2004), on these set of points $\mathcal{C}$. Let $c_r$ denote the constant approximation factor incurred by the RAM algorithm of Mettu and Plaxton (2004) which, on a set of $n$ points takes $O(n^2)$ time. The total runtime therefore increases by $O(k^2 \log(n/k)^2)$, however in the regime when $k = O(\sqrt{n})$, we get the same overall runtime of $O(n(q(k \log(n/k)) + u(k \log(n/k)))$. We state this below as a simple corollary.

**Corollary 2.** *Using a $c_r$ approximate RAM algorithm after Algorithm 1 gives a set $\bar{\mathcal{C}}$ of size $k$ such that $\mathsf{COST}(P, \bar{\mathcal{C}}) \leq \frac{\alpha\beta\gamma c_r}{\beta - \alpha} \mathsf{OPT}(P, k)$ with probability $\geq 1 - \delta$. The algorithm takes $O(n(q(k\log(n/k) + u(k\log(n/k)) + (k\log(n/k))^2)$ time and uses $s(k\log(n/k))$ space.*

## 4 Building a Coreset

In this section, we show how two popular coreset construction algorithms, in particular, of Chen (2009) and Braverman et al. (2016), which have optimal runtime (using distance oracles), can be adapted to the ANN setting. Our results show that we still get improvements in the running time of these algorithms.

### 4.1 Coreset construction I (*Algorithm 2*)

**Offline Coreset.** We first discuss how the offline coreset construction algorithm of Chen (2009) can be adapted to the case when only ANN queries are allowed. The algorithm operates on a bicriterion solution obtained from some algorithm. We have already seen that Algorithm 1 gives us one such solution and using only ANN queries. Given $k \geq 1, \epsilon > 0$, the goal is to construct a $\epsilon$-coreset of a set $P$ of $n$ points. Let $A = \{a_1, a_2, \cdots a_m\}$ be the $(\lambda, \kappa)$-bicriterion solution which implies $\mathsf{COST}(A, P) \leq \lambda\mathsf{OPT}(P, k)$ and $|A| \leq \kappa k$. The coreset construction adapted from Chen (2009) proceeds as follows.

**1. Partitioning.** The first step is to partition the space into *rings*. For every point $p \in P$, we find an ANN in the set $A$ via an ANN-Oracle. This constructs a partition of the space into $m$ sets : $P_1, P_2, \cdots P_m$, where $P_i$ is the set of points in $P$ whose ANN returned in $A$ is $a_i$. We now further partition each of these sets into concentric circles. To do this, we first define $R := \mathsf{COST}(A, P)/\kappa n$. Note that $R \leq \mathsf{OPT}(P, k)/n$, i.e. it is a lower bound on average cost. Let $t = \lceil \log(\kappa n) \rceil$ - the maximum number of concentric partitions we can have inside each $P_i$. We use $B(a, R)$ to denote a Euclidean ball of radius $R$ centred at $a$. We now define the rings $P_{ij}$ as,

$$P_{ij} = \begin{cases} P_i \cap B(a_i, R) & j = 0 \\ P_i \cap B(a_i, 2^j R) \backslash B(a_i, 2^{j-1} R) & j = 1 \text{ to } t \end{cases}$$

The main idea is that we can calculate the indices $i$ and $j$ approximately simply from the answer to the ANN query. In particular, if $p$ and $\Delta$ are the points and distance returned by ANN oracle respectively, then $i$ is the index of the point $p$ in $A$, and $j = \lceil \log(\Delta/R) - 1 \rceil$, and we get the ring construction.

**2. Sampling.** The second step is to sample points from these rings. Define $s := \lceil \frac{c\kappa^2}{\epsilon^2} \left( k\log n + \frac{1}{\delta} \right) \rceil$. For each partition $P_{ij}$, if $|P_{ij}| \leq s$, then we pick all its points and construct $S_{ij}$. Otherwise, we draw $s$ independent and identical samples from $P_{ij}$ to construct $S_{ij}$. Furthermore, each of the points in $S_{ij}$ is assigned a weight of $|P_{ij}|/s$. We finally return $S = \cup_{i,j=1}^{m,t} S_{ij}$ which we claim is a $\epsilon$-coreset of $P$, as stated in full in Theorem 5.

**Theorem 5.** *The coreset construction of Chen (2009) run on $n$ points using a $c(\delta)$-ANN oracle outputs an $\epsilon$-coreset of size $O(k\epsilon^{-2}(k\log n) + \log(1/\delta)\log n)$ with probability $\geq 1 - \delta$. The space complexity is $O(k\epsilon^{-2}(k\log n) + \log(1/\delta)\log n + s(k\log(n/k)))$ and update time is $O(q(k\log n/k) + u(k\log n/k))$*

**Streaming Coreset.** In recent years, *merge and reduce* has become a standard technique to convert offline coresets to the streaming setting. The idea is partition the stream into equal sized blocks, and then use an offline algorithm to construct coresets of these blocks. In our case, the modifed algorithm of (Chen, 2009) with ANN is used for the same. We then keep merging and reducing (in size) these coresets as the stream progresses to get a streaming coreset. The reader is encouraged to see Chen (2009) for details. The proof follows from the observation that coresets are closed under operations like set union and composition (see Proposition 1). We therefore obtain the following result.

**Theorem 6.** *The coreset construction of Chen (2009) executed on a stream of $n$ points using a $c(\delta)$-ANN oracle outputs an $\epsilon$-coreset with probability at least $1 - \delta$. The size of the coreset output is $O(k\epsilon^{-2}(k\log^8 n))$, total space complexity of the algorithm is $O(k\epsilon^{-2}(k\log^8 n) + s(k\log(n/k)))$ and the update time of the algorithm is $O(q(k\log n/k) + u(k\log n/k))$.*

### 4.2   Coreset Construction II (*Algorithm 3*)

We now discuss how another streaming coreset construction technique Braverman et al. (2016) can also be adapted to work in the ANN setting. Given a candidate solution set $C$ of $k$ centers, for a point $p$, the sensitivity $s(p)$ is defined as, $s(p) = \frac{\rho(p,\mathcal{C})}{\sum_{q \in P} \rho(q,\mathcal{C})}$ i.e its distance to the nearest center in $\mathcal{C}$ divided by sum of distances of all points to their nearest centers. At a high-level, the framework presented in Braverman et al. (2016) works in two steps. In the first step, we compute a candidate solution and compute sensitivity of all the data points; in the second step, we sample with probabilities proportional to sensitivity. As in Braverman et al. (2016), we use the algorithm of Braverman et al. (2011) to compute the set $\mathcal{C}$; however, in our setting, we use our adaptation with ANN oracles (Algorithm 1). The computational bottleneck is the step where we need to compute the sensitivities and we show how ANN oracles can accelerate it. We do not state the full algorithm due to space constraints and since it is essentially based on Algorithm 1. Line 8 of Algorithm 1 (page 11) in Braverman et al. (2016) involves computing the distance of the new point in the stream to the nearest point in the facility set. This step dominates in computation over all the other steps of the algorithm. We replace this by computing approximate nearest neighbour using an ANN oracle. This replaces the computational cost of $O(|\mathcal{C}|)$ by $q(|\mathcal{C}|)$ where $\mathcal{C}$ is the facility set. In the streaming setting, we have that the size of facility set $\mathcal{C}$ is $O(k \log n)$. We now state the detailed statement for Theorem 7.

**Theorem 7.** *Algorithm 1 of Braverman et al. (2016) executed on a stream of $n$ points using a $c(\delta)$-ANN oracle (line 8) outputs a $\epsilon$-coreset with probability at least $1 - \delta$. The size of the coreset is $O(\epsilon^{-2} k \log^2 n + s(k \log n))$ and the update time of the algorithm is $O(q(k \log n) + u(k \log n))$.*

The key idea in the proof of Theorem 7, the details deferred to the appendix, is to show that the sensitivity computed using ANN orcale (denoted as $\bar{s}(p)$) gets worsened only by a constant.

**Lemma 1.** *Using ANN, the sensitivities are approximated as $\bar{s}(p) \leq c(\delta)s(p)$ with probability $\geq 1 - \delta$*

An important property established by Braverman et al. (2016) is that even if we overestimate sensitivity by a constant, sampling using this still gives us a $\epsilon$-coreset. Therefore, the correctness of our modified algorithm follows simply by invoking the above result. Plugging this in the result of Braverman et al. (2016) gives us that the size of the coreset is $O(\epsilon^{-2} k \log^2 n + s(k \log n))$ and the update time is $O(q(k \log n) + u(k \log n))$.

## 5   Experiments

We run two experiments on `USCensus1990` dataset (available at Frank and Asuncion (2010)) on 200,000 points in 68 dimensions. In the first experiment (depicted in Figure 1a), we set the target number of clusters $k = 1000$ and evaluate the proposed method (Algorithm 1) with an *exact* nearest neighbour search and *three* approximate NN search methods, namely cover trees[1], kd-trees and LSH[2]. We also compare against standard and highly-optimized implementations of three other clustering methods, Lloyd's, Gaussian mixture model and Elkan's, available in sklearn python library (Pedregosa et al., 2011). Figure 1a shows the average cost of clustering against time taken for these methods. We see that cover trees achieve the same cost as exact NN search, but takes much less time, even less than (highly optimized) sklearn methods, which produce lower cost clusters. On the other hand, ANN methods kd-tree and LSH do poorly on time taken and/or the cost of clustering. To investigate further, we evaluate how these methods fare for increasing values of $k$. In principle, ANN search methods are developed with the goal to give sublinear query time, once the data structure is built. In contrast, in our application, the data structure has to updated plentiful times, as the algorithm proceeds. Therefore, potentially the overhead of update time can overwhelm the benefits obtained from the sublinear query time. In Figure 1b, we observed that, as expected, the time for clustering with exact NN grows linearly with $k$, whereas the overall time taken by LSH grows super polynomially with $k$. On the other hand, the time for cover trees grows logartithmically with $k$, as well as sklearn methods. However, the time for cover trees grow slower than sklearn methods and therefore is expected to perform even better in large(r) scale applications.

---

[1]Source: https://github.com/emanuele/PyCoverTree/
[2]Source: https://scikit-learn.org/stable/modules/neighbors.html

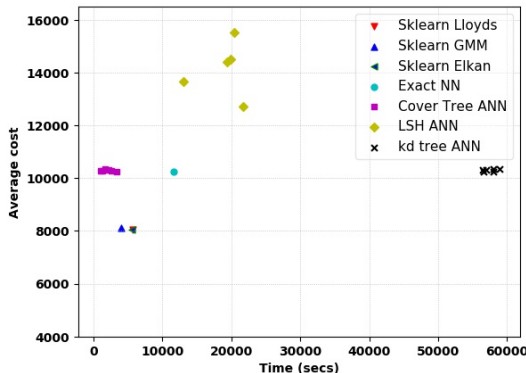

(a) Average cost of clustering against clock time for $n = 200000$ points and $k = 1000$. We also try various values of hyper-parameter in the ANN constructions, like maximum depth in cover trees, marked in the same color, in the plot.

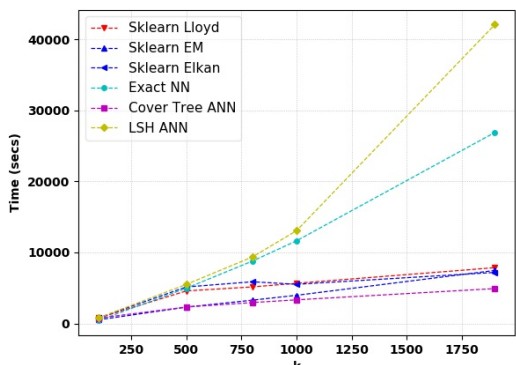

(b) Clock time against increasing values of target number of clusters $k$. We omit kd-trees here (but provide it in Figure 1 in the appendix), as the time for kd-trees grows exponentially with $k$, thereby hiding the useful comparisons otherwise.

## 6 Conclusion

We present algorithms for $k$-median clustering and coreset construction in the setting when only approximate nearest neighbour (ANN) search queries are allowed. We show, using specific ANN instantiations, that this improves upon the total runtime of previous algorithms for these problems in certain domains such as ones with constant expansion constant. We remark that our result can be extended to other clustering methods, for example $k$-means++. For the coreset construction part, we focus on algorithms of Chen (2009) and Braverman et al. (2016) since these have the optimal runtime, however as before, our results can be extended to other frameworks like Feldman and Langberg (2011).

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

# A    Additional Preliminaries

We start with giving the definition of a semi-metric space called $\alpha$-metric space which generalizes $k$-median clustering to $k$-service clustering. We first formally define $\alpha$-Approximate Triangle Inequality ($\alpha$-ATI).

**Definition 6** ($\alpha$-Approximate Triangle Inequality ($\alpha$-ATI)). *A symmetric positive-definite function* $\rho :$ $\mathcal{X} \times \mathcal{X} \to \mathbb{R}^+$ *satisfies* $\alpha$-ATI if $\forall \ x, y, z \in \mathcal{X}, \ \rho(x, z) \le \alpha(\rho(x, y) + \rho(y, z))$

In the following claim, we show that $\rho = d(\cdot, \cdot)^p$ satisfies $2^{p-1}$-ATI, where $d$ is a metric. Note that for $p = 2$, we get that $d^2$ used in $k$-means clustering satisfies 2-ATI.

**Claim 1.** *If* $(\mathcal{X}, d)$ *is a metric space, then* $\rho := d^p$ *satisfies* $\alpha$-ATI, with $\alpha = 2^{p-1}$

*Proof.* For any $x, y, z \in \mathcal{X}$, we have,

$$\rho(x, z) = d^p(x, z) \le (d(x, y) + d(y, z))^p$$
$$= 2^p \left( \frac{d(x, y) + d(y, z)}{2} \right)^p$$
$$\le 2^p \left( \frac{d^p(x, y) + d^p(y, z)}{2} \right) = 2^{p-1} \left( \rho(x, y) + \rho(y, z) \right)$$

where the last inequality follows from Jensen's inequality. □

We now define $\alpha$-metric space.

**Definition 7** ($\alpha$-metric spaces). *An* $\alpha$-metric space is a tuplet $(\mathcal{X}, \rho)$ *where* $\mathcal{X}$ *is a set endowed with a function* $\rho : \mathcal{X} \times \mathcal{X} \to \mathbb{R}^+$ *which* $\forall \ x, y, z \in \mathcal{X}$ *satisfies the following properties,*

1. $\rho(x, x) = 0$

2. $\rho(x, y) = \rho(y, x)$

3. $\rho(x, y) \le \alpha(\rho(x, z) + \rho(z, y))$

We now give an important property of coresets, called Merge and Reduce.

**Proposition 1.** *[Merge and Reduce]* *(Chen, 2009)* *Coresets satisfy the following two and properties*

1. *If* $\mathcal{S}_1$ *&* $\mathcal{S}_2$ *are* $\epsilon$-coresets of disjoint sets $\mathcal{P}_1$ *&* $\mathcal{P}_2$ respect., then $\mathcal{S}_1 \cup \mathcal{S}_2$ *is a* $\epsilon$-coreset of $\mathcal{P}_1 \cup \mathcal{P}_2$.

2. *If* $\mathcal{S}_1$ *is* $\epsilon$-coreset of $\mathcal{S}_2$ *&* $\mathcal{S}_2$ *is a* $\delta$-coreset of $\mathcal{S}_3$, then $\mathcal{S}_1$ *is a* $((1 + \epsilon)(1 + \delta) - 1)$-coreset of $\mathcal{S}_3$.

## A.1    Approximate Nearest Neighbor Oracles

The problem of nearest neighbour search is a classical geometric problem concerning with the design of data structures and algorithms which support queries for nearest neighbour search in various (metric) spaces. Nearest neighbour search has applications in diverse fields like data science, databases, computer vision etc (Bhatia et al., 2010). The typical motivation is to have query complexity *better* than an exhaustive search. Moreover, the problem is typically relaxed to return *approximate* nearest neighbours, usually in the multiplicative sense. A major challenge is to balance the tradeoff between (larger) space and (faster) query complexity.

We now review and discuss some results and techniques for approximate neighbour search in various domains. In terms of query complexity, with $n$ points in a $d$ dimensional space, we can characterize existing results into two mutually exclusive categories:

1. *ANN in low dimensions (ANN-ld)*: $\text{poly}(\tilde{d}) \cdot \text{poly} \log(n)$
2. *ANN in high dimensions (ANN-hd)*: $\text{poly}(d, n)$

where $\tilde{d}$ is some notion of intrinsic dimensionality of the space, for example: expansion constant, doubling dimension etc. We remark that there might be algorithms which do not fall in the above stated categories but for the sake for brevity, we limit ourselves to only the above two. These examples serve to instantiate the framework and therefore a detailed discussion on them is unnecessary. The interested reader is encouraged to go through the referenced material.

**ANN-ld.** In low-dimension spaces, existing works on approximate nearest neighbour search typically use *tree* data structures to give algorithms with query times logarithmic in the number of points. Some examples which fall in this category are cover trees (Beygelzimer et al., 2006), navigating nets (Krauthgamer and Lee, 2004) etc. We restrict ourselves to only *cover trees* in this article. There are also heuristics which work well in practice but their guarantees is the same as that of vanilla nearest neighbour search in the worst case. Some examples are $k$-d trees and ball trees (Bhatia et al., 2010; Bentley, 1975). To state the result of (Beygelzimer et al., 2006), we first define expansion constant below.

**Definition 8** (Expansion Constant $\tau$, (Beygelzimer et al., 2006))**.** *The expansion constant of a set $(S, \rho)$ is the smallest value of $\tau \geq 2$ such that $\left| B_\rho^S(p, 2r) \right| \leq \tau \left| B_\rho^S(p, r) \right|$ where $B_\rho^S(p, r) = B_\rho(p, r) \cap S$.*

We now restate the result of Beygelzimer et al. (2006).

**Theorem 8** ((Beygelzimer et al., 2006))**.** *Cover tree data structure is an $(1 + \epsilon)$-ANN oracle which succeeds with probability 1. The query complexity is, $q(n) = \mathcal{O}\left(\log \Delta\right) + (1/\epsilon)^{\mathcal{O}(1)}, u(n) = \mathcal{O}\left(\tau^6 \log n\right), s(n) = \mathcal{O}\left(n\right)$ where $\Delta$ is the aspect ratio of the set.*

**ANN-hd.** We now discuss the other class of results which are designed with the consideration that the dimension is large. Our discussion is primarily drawn from a survey by Andoni, Indyk and Razenshteyn (Andoni et al., 2018). In these methods, the approximate nearest neighbour search is generally reduced to its decision variant called *approximate near neighbour*, defined below.

**Definition 9** (($c, \mu, \delta$)-Approximate Near Neighbour)**.** *Given a set of $n$ points $P \subset \mathcal{X}$, build a data structure $S$ such that given any query point $q \in \mathcal{X}$, such that $P$ contains a point in $B_d(q, \mu)$, $S$ returns a point from $B_d(q, c\mu)$ with probability at least $1 - \delta$.*

The reader is encouraged to look at Har-Peled et al. (2012) to see the reduction of the approximate nearest neighbour problem to a sequence of approximate near neighbour problems. We use $(c, \mu, \delta)$ ANN oracle to refer to an oracle corresponding to the $(c, \mu, \delta)$-Approximate Near Neighbour problem. We now introduce *Locally Sensitive Hashing*, an idea central to the problem of approximate nearest neighbour search.

**Locally Sensitive Hashing (LSH).** Locally-Sensitive hashing (Indyk and Motwani, 1998) is an instantiation of the technique of randomized space partitions. The idea is to come up with a distribution of hash maps such that a query point collides *more* with points nearby, but collides with only a few points which are far away. This is formalized below.

**Definition 10** (Locally-Sensitive Hashing (Andoni et al., 2018))**.** *For a metric space $(\mathcal{X}, d)$, scale $\mu > 0$, approximation $c > 1$ and a set $P$. Then a distribution $\mathcal{H}$ over maps $h : \mathcal{X} \to P$ is called $(\mu, c\mu, p_1, p_2)$-sensitive if the following holds for any $x, y \in \mathcal{X}$*

- *If $d(x, y) \leq \mu$ then $\underset{h}{\mathbb{P}}\left[h(x) = h(y)\right] \geq p_1$*
- *If $d(x, y) > c\mu$ then $\underset{h}{\mathbb{P}}\left[h(x) = h(y)\right] \leq p_2$*

*The distribution $\mathcal{H}$ is called an LSH family and has quality $\nu = \nu(\mathcal{H}) = \frac{\log(1/p_1)}{\log(1/p_2)}$.*

We now restate a theorem from Andoni et al. (2018) which formally describes how existence of an LSH family facilitates construction of ANN oracles.

**Theorem 9** (Andoni et al. (2018))**.** *Suppose that a metric space $(\mathcal{X}, d)$ admits $(\mu, c\mu, p_1, p_2)$-sensitive LSH family $\mathcal{H}$ with scale $\mu$. Moreover, suppose that the space required to store $h(\cdot)$ is $\sigma$ and for a given $x, y \in \mathcal{X}$, $h(x)$ can be computed in $\tau$ time and $d(x, y)$ be computed in $\mathcal{O}\left(\tau\right)$ time. Let $\varrho = \varrho(\mathcal{H}) = \frac{\log(1/p_1)}{\log(1/p_2)}$. Then there exists and $(c, \mu, \delta)$ ANN Oracle with $q(n) = \mathcal{O}\left(\frac{n^\varrho \tau \log_{1/p_2} n}{p_1}\right)$ and space complexity $= \mathcal{O}\left(\frac{n^{1+\varrho}}{p_1} + \frac{n^\varrho \sigma \log_{1/p_2} n}{p_1}\right)$*

LSH is a data-independent space partitioning approach. There are other approaches which constructs data structures based on partitions that depend on the dataset. These usually improve over the results obtained via LSH. Table 2, drawn from (Andoni et al., 2018), lists some results under various metrics and methods. The query and space complexity in each of these is $\mathcal{O}\left(n^{\varrho}d\right)$ and $\mathcal{O}\left(n^{\varrho+1} + nd\right)$ respectively.

| Metric | Type | Exponent |
|---|---|---|
| $l_1$ | LSH | $\varrho = 1/c$ $\varrho \geq 1/c - o(1)$ |
| | Data-dependent hashing | $\varrho = \frac{1}{2c-1} + o(1)$ $\varrho = \frac{1}{2c-1} - o(1)$ |
| $l_2$ | LSH | $\varrho \leq 1/c^2$ $\varrho = 1/c^2 + o(1)$ $\varrho \geq 1/c^2 - o(1)$ |
| | Data-dependent hashing | $\varrho = \frac{1}{2c^2-1} + o(1)$ $\varrho = \frac{1}{2c^2-1} - o(1)$ |

Table 2: ANN search using LSH and data-dependent partitioning where $c$ is the constant in approximation

## B  Streaming $k$-median Clustering

We first give a theorem for Online Facility Location (OFL) with approximate nearest neighbour oracles. The algorithm for $k$-service clustering (Algorithm 1) uses OFL as a subroutine and therefore its guarantees are crucially dependent on the guarantees of OFL. We restate the theorem.

**Theorem 10.** *(**Restatement of Theorem 3**) Given an $c(\delta)-$ANN oracle with $u(\cdot)$ update, $q(\cdot)$ query and $s(\cdot)$ space complexity, OFL run on a stream of $n$ points from an $\alpha$-metric space with facility cost $f = \frac{L}{1+\log n/k}$ where $L \leq$ OPT gives an expected service cost of at most $3c(\delta)(\alpha+1)$OPT and the expected number of facilities produced is at most $(3c(\delta)\alpha + 1)k(1 + \log(n/k))$ conditioned on the randomness of the ANN oracle which succeeds with probability at least $1 - \delta$. Furthermore, with probability at least $1 - \delta$, the service cost is at most $\left(3c(\delta/4n)\alpha + \frac{e}{e-1}\left(1 + \log\left(\frac{4}{\delta}\right)\right)\right)$ OPT and the number of facilities produced is at most $(1 + (1 + 6\log(4/\delta)c(\delta/4n)\alpha)k(1 + \log(n/k))$. The algorithm runs in $\mathcal{O}\left(n(q(k\log(n/k)) + u(k\log(n/k)))\right)$ time.*

*Proof.* Let OPT $= \min_{K \subseteq \mathcal{X}, |K|=k} \sum_{x \in X} \min_{y \in K} \rho(x, y)$ be the optimal service cost and $\mathcal{C}^*$ be the set of facilities allocated by OPT. Let $c_i^*$'s denote the optimum $k$ facilities, and $C_i^*$'s denote the points allocated to $c_i^*$ by OPT, $i \in [k]$. Let $A_i^* = \sum_{x \in C_i^*} \rho(x, c_i^*)$ be the cost of region $C_i$, and $a_i^* = \frac{A_i^*}{|C_i^*|}$ be its average cost. Let $S_i^1$ be the first *ring* around $c_i^*$ which contains half the nearest points in $C_i^*$. Formally, $S_i^1 = \min_{K, |K|=|C_i|/2} \sum_{x \in C_i^*} \rho(x, c_i^*)$. The subsequent rings, $S_i^j = \left\{ x \in C_i : \rho(y, c_i^*) \leq \rho(x, c_i^*) \; \forall \; y \in S_i^{j-1} \right\}$ and $|S_i^j| = \frac{|C_i^*|}{2^j}$. We remark that $S_i^j$ might not be uniquely identifiable but for the sake of analysis, we only care about their existence. Let $A_i^j = \sum_{x \in S_i^j} \rho(x, c_i^*)$ be the cost of region $S_i^j$. For a point $p$, let $\rho_p^*$ denote its optimal service cost and let $\rho_p$ denote the *service cost* incurred by $p$ by our algorithm. For a region $Z \subseteq Y$, COST$(Z)$ denotes the service cost incurred by the points in region $Z$. Let COST denote the service cost incurred by our algorithm and $\mathcal{C}$ denote the set of facilities produced by our algorithm. Part 1 and 2 of the theorem follows from Lemma 2 and 3 respectively. $\square$

**Lemma 2** (OFL | Bounds in Expectation)**.** *After observing $n$ points in the stream, conditioned on the randomness of the ANN oracle, the expected service cost $\leq (1+3c(\delta_1)\alpha)$OPT and expected number of facilities $\leq 3\alpha c(\delta_1)$OPT where $\delta_1$ is the failure probability of the ANN oracle.*

*Proof.* We start with showing the bound on expected cost by analyzing two cases. We first look at the case wherein every region has a facility. Consider a region $S_i^j$ and let $q$ be the facility opened in the region.

For a subsequent point $p$, in the worst case $q$ is be the nearest facility and the service cost incurred is at most $c(\delta_1)\rho(p,q)$ with probability at least $1-\delta_1$. So we have $\rho_p \leq c(\delta_1)\alpha\left(\rho_q^* + \rho_p^*\right)$ by $\alpha$-approximate triangle inequality. By construction of $S_i^j$, $\rho_q^* \leq \rho_z^* \ \forall \ z \in S_i^{j+1}$. Summing over all the $z$ in $S_i^{j+1}$, we get $\rho_q^* \leq \dfrac{A_i^{j+1}}{\left|S_i^{j+1}\right|}$. Furthermore, summing over all the points $p$ in $S_i^j$ and doing a union bound over the ANN oracle calls with failure probability $\delta_1$ for all points in $S_i^j$, we get $\mathsf{COST}(S_i^j) \leq c(\delta_1)\alpha\left(A_i^j + \dfrac{\left|S_i^j\right|A_i^{j+1}}{\left|S_i^{j+1}\right|}\right) = c(\delta_1)\alpha\left(A_i^j + 2A_i^{j+1}\right)$ with probability at least $1-|S_i^j|\delta_1$. Summing over $j$ and doing a union bound over ANN oracle calls with failure probability $\delta_1$ for all points in $\mathcal{C}_i^*$ we get $cost(\mathcal{C}_i^*) \leq 3c(\delta_1)\alpha A_i$ with probability at least $1-|\mathcal{C}_i^*|\delta_1$. Finally, summing over $i$'s and doing a union bound over ANN oracle calls with failure probability $\delta_1$ for all points in the stream, we get that ANN oracle succeeds with probability at least $1-\delta_1$ for all points in the stream and the total (expected) cost after facilities opened in all the regions: $cost \leq 3c(\delta_1)\alpha OPT$ conditioned on the randomness of the ANN oracle.

The second case is when no region has a facility opened. For a region $S_i^j$, the expected service cost incurred before a facility is opened $\leq f$ (See Fact 1, Lang (2017)). By Lemma 9, we have that for $n$ points, and $k$ facilities, the total number of regions $j$ is less than or equal to $k\left(1 + \log(n/k)\right)$. Hence the expected total service cost before facilities opened is less than or equal to $k\left(1 + \log(n/k)\right)f$.

Combining the two cases above, expected total service cost $\leq k(1 + \log\frac{n}{k})f + 3c(\delta_1)\alpha\mathsf{OPT}$ with probability at least $1-n\delta_1$. If $f = \dfrac{L}{(1 + \log(n/k))}$ and $L \leq \mathsf{OPT}$, we get expected total service cost $\leq (1+3c(\delta_1)\alpha)\mathsf{OPT}$.

We now look at the expected facility count. After $k(1+\log n)$ facilities (one in each region), each subsequent point $p$ has a probability of $\dfrac{\rho_p}{f}$ to open a new facility. Hence the expected number of facilities $= \sum_p\left(\dfrac{\rho_p}{f}\right)$. We already showed that $\sum_p \rho_p \leq 3c(\delta_1)\alpha\mathsf{OPT}$. Therefore, we get $\mathbb{E}\left[\text{number of facilities}\right] \leq \dfrac{3c(\delta_1)\alpha\mathsf{OPT}}{f}$. $\qquad\square$

We now present high probability versions of the above bounds.

**Lemma 3** (OFL | High probability Bounds). *After observing $n$ points in the stream, with probability at least $1 - \delta$, $\mathsf{COST} \leq \left(3c(\delta/4n)\alpha + \frac{e}{e-1}\left(1 + \log\left(\frac{4}{\delta}\right)\right)\right)\mathsf{OPT}$ and the number of facilities is at most $(1 + 6\log(4/\delta)c(\delta/4n)\alpha)k(1 + \log(n/k))$.*

*Proof.* We first look at the service cost. Let $P[x,y]$ be the probability that given $x$ regions which do not yet have a facility, the remaining service cost due to points in these regions arriving prior to the region having a facility is more than $yf$. Using induction, we show that $P[x,y] \leq e^{x-y\left(\frac{e-1}{e}\right)}$. The base case $x = 0$ holds for $y \leq \frac{xe}{e-1}$. Now assume that it holds for $x < x_0$ and $y < y_0$. If the first request in one of the facility-less regions incurs a service cost of $\bar{\rho} > 0$, then

$$P[x_0,y_0] = \frac{\bar{\rho}}{f}P[x_0-1,y_0] + \left(1 - \frac{\bar{\rho}}{f}\right)P[x_0,y_0 - \frac{\bar{\rho}}{f}]$$
$$\leq \frac{\bar{\rho}}{f}e^{x_0-1-y_0\left(\frac{e-1}{e}\right)} + \left(1 - \frac{\bar{\rho}}{f}\right)e^{x_0-\left(y_0-\frac{\bar{\rho}}{f}\right)\left(\frac{e}{e-1}\right)}$$
$$\leq e^{x_0-y_0\left(\frac{e}{e-1}\right)}\left(\frac{\bar{\rho}}{ef} + \left(1 - \frac{\bar{\rho}}{f}\right)e^{\frac{\bar{\rho}e}{(e-1)f}}\right)$$
$$\leq e^{x_0-y_0\left(\frac{e}{e-1}\right)}$$

The last inequality follows because $\frac{\bar{\rho}}{ef} + \left(1 - \frac{\bar{\rho}}{f}\right) e^{\frac{\bar{\rho}e}{(e-1)f}} \leq 1 \ \forall \ \bar{\rho} \leq f$. For the total number of regions to be at most equal to $k\left(1 + \log\left(\frac{n}{k}\right)\right)$ and a failure probability of $\delta_2$, we have $e^{x-y\left(\frac{e-1}{e}\right)} = \delta_2$. Therefore, we get, $y = \frac{e}{e-1}\left(k\left(1 + \log\left(\frac{n}{k}\right)\right) + \log\left(\frac{1}{\delta_2}\right)\right)$. Hence cost is at most $\frac{e}{e-1}\left(k\left(1 + \log\left(\frac{n}{k}\right)\right) + \log\left(\frac{1}{\delta_2}\right)\right) f \leq \frac{e}{e-1}\left(\mathsf{OPT} + \log\left(\frac{1}{\delta_2}\right) f\right) \leq \frac{e}{e-1}\left(1 + \log\left(\frac{1}{\delta_2}\right)\right) \mathsf{OPT}$ with probability at least $1 - \delta_2$, when $f = \frac{L}{k(1+\log(n/k))}$. We already proved a deterministic guarantee of at most $3c(\delta_1)\alpha OPT$ with probability at least $1 - n\delta_1$ cost after all the regions have facilities. Combining it with the above bound via a union bound gives us that the total service cost is at most $\left(3c(\delta_1)\alpha + \frac{e}{e-1}\left(1 + \log\left(\frac{1}{\delta_2}\right)\right)\right) \mathsf{OPT}$ with probability at least $1 - n\delta_1 - \delta_2$.

We now look at facility count. After $k(1 + \log n)$ facilities (one in each region), each subsequent point $p$ has a probability of $\frac{\rho_p}{f}$ to open a new facility. Hence the expected number of facilities $= \mu = \sum_p \left(\frac{\rho_p}{f}\right)$. Let $A$ be the event that ANN oracle calls for all $n$ points succeed with constant factor $c(\delta_1)$. We know that this events occurs with a probability at least $1 - n\delta_1$, i.e. $\mathbb{P}[A] \geq 1 - n\delta_1$. Conditioned on event $A$, we already showed in Lemma 2 that $\sum_p \rho_p \leq 3c(\delta_1)\alpha OPT$. Therefore $\mu \leq \frac{3c(\delta_1)\alpha OPT}{f} = \mu_u$. Let the event $B_\epsilon = \{$number of facilities $\geq (1 + \epsilon)\mu_u\}$. From Chernoff bound on Bernoulli trials, we have $\mathbb{P}[B'_\epsilon|A] \leq \exp\left(-\frac{\mu_u \epsilon^2}{3}\right) \leq \delta_2$. Solving for $\epsilon$, we get $\epsilon \geq \sqrt{\frac{3\log(1/\delta_2)}{\mu_u}}$. Therefore, conditioned on event $A$, the number of facilities $\leq (\mu_u + \sqrt{3\log(1/\delta_2)\mu_u}) \leq \mu_u(1 + 3\log(1/\delta_2)) \leq (1 + 3\log(1/\delta_2))c(\delta_1)\alpha k\left(1 + \log\frac{n}{k}\right)\frac{\mathsf{OPT}}{L}$ when $f = \dfrac{L}{k\left(1 + \log\dfrac{n}{k}\right)}$ with probability at least $1 - \delta_2$. Moreover, since $\mathbb{P}[B_\epsilon] \geq \mathbb{P}[B_\epsilon|A] \cdot \mathbb{P}[A] \geq (1 - n\delta_1)(1 - \delta_2) \geq 1 - (n\delta_1 + \delta_2)$, therefore, combining the two cases, we have the number of facilities $\leq (1 + 6\log(1/\delta_2)c(\delta_1)\alpha)k\left(1 + \log\frac{n}{k}\right)\frac{\mathsf{OPT}}{L}$ with probability at least $1 - (n\delta_1 + \delta_2)$. Setting $\delta_1 = \frac{\delta}{4n}$ and $\delta_2 = \frac{\delta}{4}$ and using a union bound over the above guarantees of cost and number of facilities finishes the proof. $\qquad\square$

**Theorem 11.** *(**Restatement of Theorem** 4) Given a set of $n$ points $P$ from an $\alpha$-metric space $(\mathcal{X}, \rho)$ observed in a stream and a positive integer $k$, Algorithm 1 executed with $\beta = 2\alpha^2 c_{OFL} + 2\alpha$ and $\gamma = \max\left\{4\alpha^3 c_{OFL}^2, \beta k_{OFL} + 1\right\}$, where $c_{OFL} := \left(3c(\delta/4n)\alpha + \frac{e}{e-1}\left(1 + \log\left(\frac{4}{\delta}\right)\right)\right), k_{OFL} := (1 + 6\log(1/\delta_2)c(\delta_1)\alpha)$ returns a set $\mathcal{C}$ with $|C| = \mathcal{O}\left(k\log(n/k)\right)$ such that $\mathsf{COST}(P, \mathcal{C}) \leq \frac{\alpha\beta\gamma}{\beta - \gamma}\mathsf{OPT}(P, k)$ with probability at least $1 - \delta$. The algorithm runs in time $\mathcal{O}\left(n \cdot q(k\log(n/k))\right)$ and takes space $\mathcal{O}\left(k\log(n/k)\right)$.*

*Proof.* From Theorem 3, we get that at every phase, OFL produces a bicriterion approximation to our problem with high probability. Note that $c_{OFL} := \left(3c(\delta/4n)\alpha + \frac{e}{e-1}\left(1 + \log\left(\frac{4}{\delta}\right)\right)\right)$ is the constant approximation factor of the service cost of OFL in the high probability bound. Similarly, $k_{OFL} := (1 + \sqrt{3c(\delta/4n)\alpha\log(4/\delta)})$ such that OFL opens at most $k_{OFL}k\left(1 + \log n\right)\frac{\mathsf{OPT}}{L}$ facilities with high probability.

We remind the reader that the algorithm proceeds in phases, wherein at each phase we guess lower bound to the optimal cost and start reading points from the stream. If the guess is too low, we either end up opening too many facilities or the service cost grows large. At this point, we trigger a phase change and enter the next phase in the algorithm where we increase our guess to the lower bound by a constant factor of $\beta$, and first read the already clustered points and then start reading the remaining ones. If we keep incrementing the phase, at some point our guess to the lower bound exceeds the optimal cost. We call this phase the *critical phase*, formally defined as $q = \min\{i : \beta L_i > \mathsf{OPT}\}$. From Lemma 4, we have that with probability at least $1 - \delta$, the algorithm exits before or at the critical phase.

For a point $x \in X$, let $c_x^i$ denote its representative in phase $i$. Let $j$ be the phase when $x$ was first clustered. By $\alpha$-ATI of $\rho$, we have,

$$\rho(x, c_x^l) \leq \sum_{i=1}^{l-j} \alpha^i \rho(c_x^{l-i}, c_x^{l-i+1}) + \alpha^{l-j} \rho(x, c_x^j)$$

Summing over all points $x \in X$, with probability at least $1 - \delta$ of terminating before the critical phase, we have,

$$\begin{aligned}
\mathsf{COST} &\leq \sum_{i=1}^{l-j} \alpha^i \sum_x \rho(c_x^{l-i}, c_x^{l-i+1}) + \alpha^{l-j} \sum_x \rho(x, c_x^j) \\
&\leq \sum_{i=1}^{l} \alpha^i \gamma L_{l-i} = \gamma L_l \sum_{i=1}^{l} \left(\frac{\alpha}{\beta}\right)^i \\
&= \frac{\alpha \gamma L_l}{\beta - \alpha} \leq \frac{\alpha \beta \gamma}{\beta - \alpha} \mathsf{OPT}
\end{aligned}$$

where in the second equality we used the fact that the cost in phase $i$ is upper bounded by $\gamma L_i$. Note that our value of $\beta$ ensures $\alpha < \beta$, so we can use the sum of an infinite geometric progression. In the third inequality, we used that $L_i = \frac{L_l}{\beta^{l-i}}$. In the last inequality, we use $L_l \leq \mathsf{OPT}$ with probability at least $1 - \delta$, since from Lemma 4 we know that the algorithm terminates before or at the critical phase. $\qquad \square$

We show below that before or at the critical phase, the algorithm finishes and produces a bicreterion approximate solution.

**Lemma 4.** *With probability at least the success probability of online facility location, Algorithm 1 terminates at or before the critical phase.*

*Proof.* Let $X_l$ be the stream in phase $l$. We first show that cost of clustering $X_l$ is $\leq \alpha \mathsf{OPT} + \gamma \left(\frac{\alpha^2}{\beta - \alpha}\right) L_l$.

Consider a point $x \in X$. We remind the reader that $c_x^i$ represents $x$ in phase $i$. Let $j$ be the phase in which $x$ gets clustered the first time. Note that $c_x^i = x \ \forall i < j$. Moreover, once $x$ gets clustered, gets represented by some point not necessarily $c_x$ in all subsequent phases. Let $c_{x,i}^*$ denote the optimal facility fro representation of $x$ in phase $i$. We have that in phase $l$, the optimal service cost of $x$ is at most,

$$\begin{aligned}
\rho(c_x^{l-1}, c_{x,l-1}^*) &\leq \alpha(\rho(x, c_{x,l-1}^*) + \rho(c_x^{l-1}, x)) \\
&\leq \alpha \rho(x, c_{x,l-1}^*) + \sum_{i=2}^{l-j} \alpha^i \rho(c_x^{l-i}, c_x^{l-i+1}) \\
&\quad + \alpha^{l-j} \rho(x, c_x^j)
\end{aligned}$$

where we used $\alpha$-ATI of $\rho$ to expand the cost incurred in phase $l$ into the costs across previous phases. Let $\mathsf{OPT}_i$ denote the optimal cost of clustering at phase $i$. Summing over all points $x$ in the stream, we get

$$\mathsf{OPT}_l \leq \sum_x \rho(c_x^{l-1}, c_{x,l-1}^*)$$

$$\leq \alpha \sum_x \rho(x, c_{x,l-1}^*) + \sum_{i=2}^{l-j} \alpha^i \sum_x \rho(c_x^{l-i}, c_x^{l-i+1})$$

$$+ \alpha^{l-j} \sum_x \rho(x, c_x^j)$$

$$\leq \alpha\mathsf{OPT} + \sum_{i=2}^{l-j} \alpha^i \gamma L_{l-i} + \alpha^{l-j}\gamma L_j$$

$$\leq \alpha\mathsf{OPT} + \alpha\gamma \sum_{i=1}^{l-1} \alpha^{l-i} L_i$$

$$\leq \alpha\mathsf{OPT} + \alpha\gamma \sum_{i=1}^{l-1} \left(\frac{\alpha}{\beta}\right)^{l-i} L_l$$

$$\leq \alpha\mathsf{OPT} + \frac{\alpha^2\gamma}{\beta - \alpha} L_l$$

where in the second and third inequality we used that the service cost in phase $i$ is bounded by $\gamma L_i$, and $L_i = \frac{L_l}{\beta^{l-i}}$. Therefore, we see that if we ensure that $L_l \leq \mathsf{OPT}$, we get a constant approximate solution. Note that as earlier defined, critical phase is the first phase which violates this. We now show that with high probability the algorithm terminates before or at the critical phase. Let $q$ be the critical phase, therefore $\beta L_q \geq \mathsf{OPT}$ . We have,

$$\mathsf{OPT}_q \leq \alpha\mathsf{OPT} + \frac{\alpha^2\gamma}{\beta - \alpha} L_q$$

$$\leq \left(\alpha\beta + \frac{\alpha^2\gamma}{\beta - \alpha}\right) L_q$$

By Theorem 3, we have that in phase $q$, OFL guarantees a bicriterion approximate solution such that $\mathsf{COST}_q \leq c_{OFL}\mathsf{OPT}_q$ and the number facilities at most $\beta k_{OFL}(1 + \log{(n/k)})\frac{\mathsf{OPT}}{L}$. We define $\beta := 2\alpha^2 c_{OFL} + 2\alpha, \gamma := \max\left\{4\alpha^3 c_{OFL}^2, \beta k_{OFL} + 1\right\}$. We therefore get

$$\mathsf{COST} \leq c_{OFL}\left(\alpha(2\alpha^2 c_{OFL} + 2\alpha) + \frac{\alpha^2\gamma}{2(2\alpha^2 c_{OFL} + \alpha)}\right) L_q$$

$$\leq \gamma L_q$$

Moreover, the number of facilities is $\leq \beta k_{OFL}(1 + \log{(n/k)})\frac{\mathsf{OPT}}{L} \leq (\gamma - 1)k(1 + \log(n/k))$. Therefore, the algorithm doesn;t proceed to the next phase. $\qquad\square$

## C    Building a coreset

**Theorem 12. (*Reminder of Theorem* 5)** *The coreset construction algorithm of* Chen (2009) *executed on $n$ points using a $c(\delta)$-ANN oracle outputs a $\epsilon$-coreset with probability at least $1 - \delta$. The size of the coreset output is $\mathcal{O}\left(k\epsilon^{-2}(k\log n) + \log(1/\delta)\log n\right)$, total space complexity is $\mathcal{O}\left(k\epsilon^{-2}(k\log n) + \log(1/\delta)\log n + s(k\log(n/k))\right)$ and the update time of the algorithm is $\mathcal{O}\left(q(k\log n/k) + u(k\log n/k)\right)$.*

*Proof.* Lemma 5 gives us the cost guarantee. We now only need to look at size and update times. The size of the coreset is $|S| = mts = \mathcal{O}\left(k\epsilon^{-2}(k\log n) + \log(1/\delta)\log n\right)$. Note that if we use our algo-

rithm to get the bicriterion solution, its size is $\mathcal{O}\left(k\log(n/k)\right)$. Therefore the total space complexity is $\mathcal{O}\left(k\epsilon^{-2}(k\log n) + \log(1/\delta)\log n + s(k\log(n/k))\right)$. The update time also follows from the above. $\qquad\square$

**Lemma 5.** *For all $C \subseteq P$ such that $|C| \leq k$, with probability at least $1 - \delta$, we have,*

$$|\mathsf{COST}(C, P) - \mathsf{COST}(C, S)| \leq \epsilon \; \mathsf{COST}(C, P)$$

*Proof.* For a fixed $C$, from Lemma 6, with $\xi = \frac{\epsilon}{2(2c(\delta_1)+1)\kappa}$ and $\delta_2 > 0$, with probability at least $1 - \delta_2$, we get,

$$|\mathsf{COST}(C, P_{i,j}) - \mathsf{COST}(C, S_{i,j})|$$
$$\leq \frac{\epsilon}{2(2c(\delta_1) + 1)\kappa} |P_{i,j}| \operatorname{diam}(P_{i,j})$$

Moreover, from triangle inequality, we have,

$$|\mathsf{COST}(C, P) - \mathsf{COST}(C, S)|$$
$$\leq \sum_{i,j} |\mathsf{COST}(C, P_{i,j}) - \mathsf{COST}(C, S_{i,j})|$$
$$\leq \frac{\epsilon}{2(2c(\delta_1) + 1)\kappa} \sum_{i,j} |P_{i,j}| \operatorname{diam}(P_{i,j})$$

For a point $p \in P_{ij}, j \geq 1$, note that $2^{j-1}R \leq \tilde{\rho}(p, A) \leq c(\delta_1)\rho(p, A)$ with probability at least $1 - \delta_1$. Therefore, $2^j R \leq \max(R, 2c(\delta_1)\rho(p, A)) \leq R + 2c(\delta_1)\rho(p, A)$ for $j \geq 0$. Taking a union bound over all points, we get that with probability at least $1 - n\delta_1$,

$$\sum_{i,j} |P_{i,j}| 2^j R = \sum_{i,j} \sum_{p \in P_{i,j}} (2c(\delta_1)\rho(p, A) + R)$$
$$= \sum_{p \in P} (2c(\delta_1)\rho(p, A) + R)$$
$$= 2c(\delta_1)\mathsf{COST}(A, P) + |P| R$$
$$= 2c(\delta_1)\mathsf{COST}(A, P) + nR$$
$$\leq (2c(\delta_1) + 1)\mathsf{COST}(A, P)$$
$$\leq (2c(\delta_1) + 1)\kappa\mathsf{OPT}(P, k)$$

where the last inequality follows because $R = \frac{\mathsf{COST}(A,P)}{\kappa n} \leq \frac{\mathsf{OPT}(P,k)}{n}$. Moreover, $\operatorname{diam}(P_{i,j}) \leq 2 \cdot 2^j R$. We therefore get, with probability at least $1 - n.\delta_1 - m.(t+1)\delta_2$,

$$|\mathsf{COST}(C, P) - \mathsf{COST}(C, S)|$$
$$\leq \frac{\epsilon}{2(2c(\delta_1) + 1)\kappa} 2.(2c(\delta_1) + 1)\kappa\mathsf{OPT}(P, k) \leq \epsilon\mathsf{COST}(C, P)$$

Setting $\delta_1 = n^{-k}\delta/2$ and $\delta_2 = n^{-k}\delta/2m(t+1)$, and noting that there can be $n^k$ ways of selecting $k$ points from $P$, we get that with probability at least $1 - \delta$, for all $C$ of size at most $k$,

$$|\mathsf{COST}(C, P) - \mathsf{COST}(C, S)| \leq \epsilon\mathsf{COST}(C, P)$$

$\qquad\square$

**Lemma 6.** *Let $V$ be a a set of points from a metric space $(\mathcal{X}, d)$ and let $\delta', \xi > 0$. Let $U$ be a set constructed by sampling $s' = \lceil \xi^{-2} \log(2/\delta') \rceil$ points independently and identically from $V$, and each point is assigned weight $|V|/|U|$. For a fixed set $C \subseteq \mathcal{X}$, the following holds with probability at least $1 - \delta'$*

$$|\mathsf{COST}(C, V) - \mathsf{COST}(C, U)| \leq \xi|V|diam(V)$$

Proof in (Chen, 2009) which crucially uses Lemma 7 stated below.

**Lemma 7.** *Let $M \geq 0$ and $\eta$ be fixed constants and let $h(\cdot)$ be a function defined on a set $V$ such that $\eta \leq h(v) \leq \eta + M$. Let $U$ be a set of $s \geq \frac{M^2}{2\delta^2} \log(2/\delta)$ samples drawn independently and identically from $V$. Let $h(U) = \sum_{u \in U} h(u)$ and $h(V) = \sum_{v \in V} h(v)$. For $\delta > 0$, we then have,*

$$\mathbb{P}\left[\left|\frac{h(V)}{|V|} - \frac{h(U)}{|U|}\right| \geq \delta\right] \leq \lambda$$

**Theorem 13.** *(**Restatement of Theorem 6**) Algorithm of Chen (2009) executed on a stream of $n$ points using a $c(\delta)$-ANN oracle (line 8) outputs a $\epsilon$-coreset with probability at least $1 - \delta$. The size of the coreset output is $\mathcal{O}\left(\epsilon^{-2}k \log^2 n + s(k \log n)\right)$ and the update time of the algorithm is $\mathcal{O}\left(q(k \log n) + u(k \log n)\right)$.*

*Proof.* The proof is the same as that of merge and reduce in Chen (2009) □

**Theorem 14.** *(**Restatement of Theorem 7**) Algorithm 1 of Braverman et al. (2016) executed on a stream of $n$ points using a $c(\delta)$-ANN oracle (line 8) outputs a $\epsilon$-coreset with probability at least $1 - \delta$. The size of the coreset output is $\mathcal{O}\left(\epsilon^{-2}k \log^2 n + s(k \log n)\right)$ and the update time of the algorithm is $\mathcal{O}\left(q(k \log n) + u(k \log n)\right)$.*

*Proof.* We will show that using ANN oracle, the sensitivity of a point increases with at worse a constant factor.

Let $p$ be the point observed in the stream and let $\bar{s}(p)$ denote its sensitivity computed using ANN oracle. We show in the following claim that this worsens the sensitivity by a constant factor. In claim 8, we show that $\bar{s}(p) \leq c(\delta) s(p)$ with probability $\geq 1 - \delta$.

We now carefully invoke the result of Braverman et al. (2016) that overestimating sensitivity by a constant factor still produces a $\epsilon$-coreset. In particular, we set probability of failure of the oracle $\delta'$ as $\delta' = \delta/2n$ and condition on the event that it succeeds for each point. This establishes that we get a $\epsilon$-coreset with probability at least $1 - \delta$.

Since the ANN data structure takes $s(k \log n)$, the size of the coreset is $\mathcal{O}\left(\epsilon^{-2}k \log^2 n + s(k \log n)\right)$, where $\mathcal{O}\left(\epsilon^{-2}k \log^2 n\right)$ is the size of coreset obtained by Braverman et al. (2016). The update time guarantee simply follows query and update time complexities of the ANN data structure. □

**Lemma 8.** *The sensitivities are approximated as $\bar{s}(p) \leq c(\delta) s(p)$ with probability $\geq 1 - \delta$.*

*Proof.* For a point $q$, let $a$ be its nearest neighbour in $\mathcal{C}$ and let $\bar{a} \in \mathcal{C}$ be the point output by ANN oracle. Let $\bar{\rho}(q, \mathcal{C})$ denote the distance of $q$ to the point output by ANN oracle, i.e. $\bar{\rho}(q, \mathcal{C}) = \rho(q, \bar{a})$. From the definition of ANN oracle, we have that $\rho(q, a) \leq \rho(q, \bar{a}) \leq c(\delta)\rho(q, a)$ with probability at least $1 - \delta$. We use this to upper and lower bound the numerator and denominator of sensitivity as follows.

$$\bar{s}(p) = \frac{\bar{\rho}(p, \mathcal{C})}{\sum_{q \in P} \bar{\rho}(q, \mathcal{C})} \leq c(\delta)\frac{\rho(p, \mathcal{C})}{\sum_{q \in P} \rho(q, \mathcal{C})} = c(\delta)s(p)$$

□

# D   Auxiliary results

**Lemma 9.** *With $n$ data points, $k$ facilities, let $c_i$ denote the (optimal) facilities, $C_i$ the set of points assigned to facility $c_i$ $i \in [k]$. Let $S_i^j = \{x \in C_i : \rho(y, c_i) \leq \rho(x, c_i) \leq \rho(y, x_i) \ \forall \ y \in S_i^{j-1} \ and \ \forall z \in S_i^{j+1}\}$ and $|S_i^j| = \frac{|C_i|}{2^j}$. Let the number of regions $N = \left|\left\{S_i^j \ \forall \ i \in [k] \ \forall \ j\right\}\right|$. Then,*

$$(1 + \log n) \leq N \leq k\left(1 + \log\left(\frac{n}{k}\right)\right)$$

*Proof.* We proof by demonstrating when the two ends of the bound hold. The lower bound is attained if there is only 1 facility (or $k$ facilities, but all the $n$ points belong to only of them, and $k-1$ has no points). The upper bound is attained when all the $k$ facilities have equal number of points i.e. $\frac{n}{k}$.

## E   Additional plots

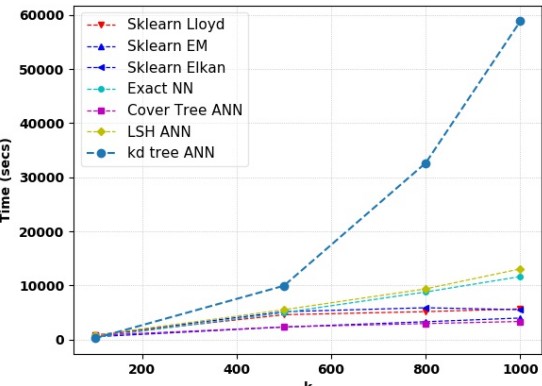

Figure 1: The plot of time taken against target number of clusters with exact and three ANN methods (kd-tree included) and sklearn methods. As is apparent, the time taken for kd tree increases exponentially with $k$, where as sklearn methods and cover tree grows sublinearly.

