# OpenReview forum: "Clustering using Approximate Nearest Neighbour Oracles"
_TMLR — Accepted by TMLR_

### Review · Reviewer_cdYj · 2022-11-16

**Summary Of Contributions:**

The paper studies the ability to create streaming coresets for k-median clustering (on n points) when the data can only (or efficiently) accessed via approximate nearest neighbor (ANN) queries.  I'm particular in spaces with "constant expansion constant" then a ANN which exploits that can reduce the runtime from O(nk log(n)) to O(n log(k log(n)).


**Audience:**

Yes

**Claims And Evidence:**

Yes

**Requested Changes:**

  - remove discussion of improving log(n) -> log (n/k).  Its mostly (entirely?) not an improvement, and it makes the analysis unnecessarily more complicated.

  - Explain the claimed results for constant expansion constant more carefully, including carefully stating the ANN results for this case, and providing a theorem.

More writing issues:
  - p5,l3 : rho(x_z) -> rho(x,z)

  - p8,l7 : "distance ruined by ANN" -> "distance returned by ANN"

  - p8, l-9 : "due to space constraints" (there are not space constraints, but the other rational for not writing it out in full detail is sufficient).


**Strengths And Weaknesses:**


Strengths:
 + the exploration of streaming clustering algorithms to exploit fast ANN algorithms is an interesting direction, and establishing that these algorithms work is worth publication at TMLR.

 + the analysis with constant expansion factor is an interesting and novel result.  The experiments with Cover Tree ANNs justify this analysis.

 + the paper is mostly written clearly and accessibly (although a few comments/suggestions below).


Weaknesses:
 - the paper emphasize improvements from log(n) to log(n/k) inside various larger bounds.  These mostly do not provide any real improvement.  Note that O(log (n/k)) = O(log n) unless k = Omega(n / polylog(n)) or so.  In this case, any factor of k = Omega(n / polylog(n)) and then other algorithms that are O(n) or O(n^2) may be preferable.

 - For instance before Corollary 1, the paper asks us to consider when k = O(sqrt{n}) and then states a runtime bound of O(n(q(k log(n/k) + u(k log(n/k)), but in this case the log(n/k) terms are O(log n).

 - Theorem 6 also advertises an algorithm with space complexity that includes a term that is O(k^2 ... + s(k log (n/k)).  Since O(n) space is trivial, then this is only interesting when k = o(sqrt{n}) and hence the log(n/k) = O(log n).


 - The claimed (in Intro) improved results for constant expansion constant are stated in the Introduction and probably implied in Table 1, but are not formally stated.

---

### Review · Reviewer_gXjA · 2022-12-09

**Summary Of Contributions:**

In this paper, the authors investigate the clustering problem in a streaming setting when only approximate nearest neighbor (ANN) search queries are allowed. They present one improved algorithm for k-median clustering and two improved methods for coreset construction. They demonstrate that by using the ANN instantiations instead of the exhaustive linear scan, the time complexity of these three algorithms can be reduced without surrendering the space complexity. Moreover, for the streaming k-median clustering algorithm, the space complexity can be further reduced.

**Audience:**

Yes

**Broader Impact Concerns:**

No.

**Claims And Evidence:**

Yes

**Requested Changes:**

Please answer the questions or fix the issues in W1-W6, especially W1, W2, and W5.

**Strengths And Weaknesses:**

Strengths:

S1. Clustering data points in a streaming setting is an important problem.

S2. They provide theoretical analyses for the proposed algorithms, and the improvement regarding the runtime and space is convincing.

S3. They conduct some experiments to validate the the runtime improvement of the proposed streaming k-median clustering method compared with previous algorithms.

Weaknesses:

W1. The authors claim that "their results are presented in a general setup wherein one has access to the geometry of the set via blackbox ANN oracles." Nonetheless, I am not very sure whether this setting is general or not. It will be more convincing if the authors can provide some actual applications for demonstration.

W2. They claim their results can be extended to other clustering methods (e.g., k-means) and M-estimators, but I fail to find more discussions in the later sections. Can they justify their claim with more discussion (and more experiments)?

W3. I suppose the $k$-median clustering is a special case of $k$-service clustering with $\alpha=1$ but not $\alpha=2$, as the $k$-median clustering is with respect to 1-norm with $p=1$.

W4. Definitions 1-4 could be improved by specifying the range of input parameters.
For example, for Definitions 1-3, it would be clearer to add $\lambda > 1$, $\kappa > 1$, and $0 < \epsilon < 1$.
Moreover, $z$ is not defined in Definition 1; I suppose it is a typo and should be replaced by $c$.
In Definition 3, $Z \subseteq X$ should be $Z \subseteq \mathcal{X}$.
In Definition 4, $x \in \mathcal{C}$ should be $x \in \mathcal{X}$.

W5. I appreciate they conduct the experiments for validation, but I still have several concerns as follows:

(1) Using a single dataset (with low intrinsic dimensions) for performance validation might be incomplete and contain some bias. Can the authors consider at least one more datasets for a further demonstration?

(2) They only validate Algorithm 1. Can they conduct more experiments to validate the improvement of Algorithms 2 and 3 for the coreset construction?

(3) The analysis of the experimental results for the highly optimized clustering methods is missing. In Figure 1(b), why does the curve of Sklearn Elkan decrease from k=750 to k=1000?

W6. The presentation can be further improved. Some typos are found as below:

(1) "... An important remark is that second framework using the $O(1)$ ..." ==> "... An important remark is that the second framework uses the $O(1)$ ..." (In the paragraph after Theorem 2)

(2) "... in constant expansions paces..." ==> "... in constant expansions spaces..." (In the **Applications** paragraph of Section 1)

(3) $\rho(x, y) \leq \alpha(\rho(x_z), \rho(z, y))$ ==> $\rho(x, y) \leq \alpha(\rho(x, z), \rho(z, y))$ (In the **$\alpha$-metric Spaces and $k$-service clustering** paragraph of Section 2).

(4) "we observe data points one at at time..." ==> "we observe data points one at a time..." (In the 1st paragraph of Section 3)

(5) "The algorithm in full in presented in Algorithm 1..." ==> "The algorithm in full is presented in Algorithm 1..." (In the 1st paragraph of Section 3)

(6) "We know discuss how another streaming coreset construction..." ==> "We now discuss how another streaming coreset construction..." (In the 1st paragraph of Section 4.2)

---

### Review · Reviewer_3X13 · 2023-01-31

**Summary Of Contributions:**

The paper considers streaming K-means and K-median clustering and focuses on the setting where the geometry can only be accessed via nearest neighbour oracles. O(1)-approximate algorithms are provided with time complexity depending on the efficiency of the underlying ANN oracle. For special cases when ANN is cheap, e.g., constant expansion space, the running time of the proposed algorithm is better than prior work.

**Audience:**

Yes

**Broader Impact Concerns:**

There is no concern on the ethical implications of the work.

**Claims And Evidence:**

Yes

**Requested Changes:**

1. The experimental studies are somewhat week. Maybe more datasets should be used, and expansion constant should be empirically estimated.
2. I think the space complexity of the algorithms are in terms of d-dimensional points. The author should be more explicit on the space complexity.
3. I suggest the authors put existing bounds on streaming clustering in Table 1 as well, so that the reader can quickly compare the results from this submission to previous ones.
4. Since the new algorithm gets improved results only for special cases, I suggest the authors add discussions on how often such special cases happen in real applications.

**Strengths And Weaknesses:**

Strengths
1. K-means and K-median clustering are fundamental and the setting considered here is interesting.
2. The work provide a systematic study on the setting, and new algorithms with theoretical guarantees are provided.
3. Non-trivial theoretical improvements are obtained for special cases. And the space complexity is also slightly improved.
4. Preliminary experimental results are provided to support the theoretical findings.

Weaknesses:
1. The experimental studies are somewhat week.
2. Most technical ingredients in the new algorithms follow ideas from previous literature, and I think the overall technical contribution is not significant.

---

### Decision · Action_Editors · 2023-03-03

**Recommendation:** Accept with minor revision

**Comment:**

This paper proposed a streaming coreset construction for k-median clustering using approximate nearest neighbor (ANN) queries.  All reviewers agreed that the problem studied in this paper is interesting and important, and theoretical analysis for the proposed algorithms deserve to be published in TMLR.  We expect that the authors will address the following issues in the final version:

- State explicitly that the improvement on the space complexity manifests only in certain restrictive regimes.
- Add a theorem for the constant expansion setting.
- "Since the reported plot does not have confidence bars, we cannot conclude whether this decrease is statistically significant or not."   Please try to add confidence bars to your plots.
- Revise and add details of existing bounds in Table 1.
- Fix all typos and minor issues mentioned by the reviewers

AC

**Audience:**

The paper studies how to use approximate nearest neighbor search for (streaming) coreset construction for k-median clustering.  Clustering and nearest neighbor search are important problems in machine learning.

**Claims And Evidence:**

All the major claims made in the submission are supported by theorems and theoretical proofs.